# Generating Dialogue Responses From A Semantic Latent Space

## Abstract

Generic responses are a known issue for open-domain dialog generation. Most current approaches model this one-to-many task as a one-to-one task, hence being unable to integrate information from multiple semantically similar valid responses of a prompt. We propose a novel dialog generation model that learns a semantic latent space, on which representations of semantically related sentences are close to each other. This latent space is learned by maximizing correlation between the features extracted from prompt and responses. Learning the pair relationship between the prompts and responses as a regression task on the latent space, instead of classification on the vocabulary using MLE loss, enables our model to view semantically related responses collectively. An additional autoencoder is trained, for recovering the full sentence from the latent space. Experimental results show that our proposed model eliminates the generic response problem, while achieving comparable or better coherence compared to baselines.

## 1 Introduction

The sequence-to-sequence (Sutskever et al., 2014) framework has become a popular choice for designing open-domain neural response generation systems (Vinyals & Le, 2015). Recently, transformer (Vaswani et al., 2017) based models have received increasing attention (Wolf et al., 2018). Those models typically involves maximizing the probability of the ground truth response given the input prompt, trained using a cross entropy loss on the 1-of-n coding vocabulary.

However, it is a known problem that models tend to generate bland and uninformative responses, such as *"I don't know"* (Serban et al., 2016; Sordoni et al., 2015), despite the presence of more specific responses in the training data than generic responses. This problem is not caused by model optimization error or decoding search error (Holtzman et al., 2019); on the contrary this is a result of a fundamental deficiency of the end-to-end maximum likelihood objective, which models the one-to-many task as a one-to-one task.

The specific issue is the following. A sequence-to-sequence decoder trained using maximum likelihood objective tends to prefer uninformative stop word with higher frequency in training data, over more informative words with lower frequency, because the model loss treat each informative word independently and ignoring their semantic relatedness. A similar effect happens on the utterance level when using beam search decoding. Beam search aims to find the most probable utterance. Each sentence is treated independently, and the model is unable to use semantic similarity between different utterances (Qiu et al., 2019). When specific responses collectively have a high probability, it is diluted by the large number of variations and possibilities of specific answers. On the other hand, generic responses have much fewer variations, thus they become the most probable response sequences. An alternative decoding method to beam search is sampling (Holtzman et al., 2019; Fan et al., 2018), which does not suffer from this problem. However, sampling does not consider the subsequent words during decoding, and the randomness in word choice makes it prone to generating implausible responses and grammatical errors.

Aiming to take into account the semantic relatedness of those diverse specific responses, we propose a novel model that learns a shared latent space with semantic information, on which semantically related sentences tend to be close to one another. The generation process could be separated into two steps. The first step predicts a sentence vector of the response on the semantic latent space. Since predicting a vector is a regression problem in the latent space instead of classification in the

vocabulary as in MLE loss, our model is able to learn that most of the probability mass of the response is around the cluster of possible specific responses. The second step constructs the full response sentence from the predicted vector.

Specifically, our semantic latent space is learned by maximizing the canonical correlation (Hotelling, 1936) between the features extracted from the prompt and the response. This objective preserves information most correlated with the other sentence in the pair. Since semantically similar responses are likely to correspond to a similar set of prompts, they will have similar representations in the latent space. Generic responses correlate with a different and much larger set of inputs than specific responses, so they will have very different representations in the latent space. This is illustrated in Figure 1 showing our model's representations of prompts and responses on a t-SNE plot.

(a)

| Prompt $D_x$ (Light red) | Prompt $D_x$ (Light green) |
| --- | --- |
| what instruments do you play ? | what do you do for a living ? |
| **Response $D_y$ (Dark red)** | **Response $D_y$ (Dark green)** |
| i practice the piano every day . 
 i can play anything on my electric violin . 
 i am learning the guitar . 
 i like the drums a lot 
 i used to play clarinet . 
 i play trombone , alto sax , baritone , and trumpet . you ? 
 my parents taught me flute | i work as an elementary teacher . 
 i am an olympic gymnast 
 aside from nursing , i work at a bar to pay for school . 
 i'm a janitor , but i also play music at night . you ? 
 since i was fired i found a job in insurance . 
 i work part time as a bartender , but i don't drink any alcohol 
 mechanical engineering is my day job . |
| **Prompt $D_x$ (Light blue)** | **Prompt $D_x$ (Light yellow)** |
| what is your favorite color ? | ya , are you a female ? |
| **Response $D_y$ (Dark blue)** | **Response $D_y$ (Dark yellow)** |
| my favorite color is green and whats yours 
 i like red too , with a bit of yellow . like a superhero ! 
 blue color makes me happy 
 mine is orange ! 
 i like rainbow colors , you ? 
 red , blue , green , and yellow . i am thinking purple too 
 strangely my favorite is grey ! | yes i am a woman . 
 not much to tell , i'm an average male . tell me about you . 
 female 
 i am just a boy with a heart outside my body 
 i am a 12 year old female |
| **Prompt $D_x$ (Brown)** | **Response $D_y$ (Black)** |
| what do you do for a living ? | I don't know . |

(b)

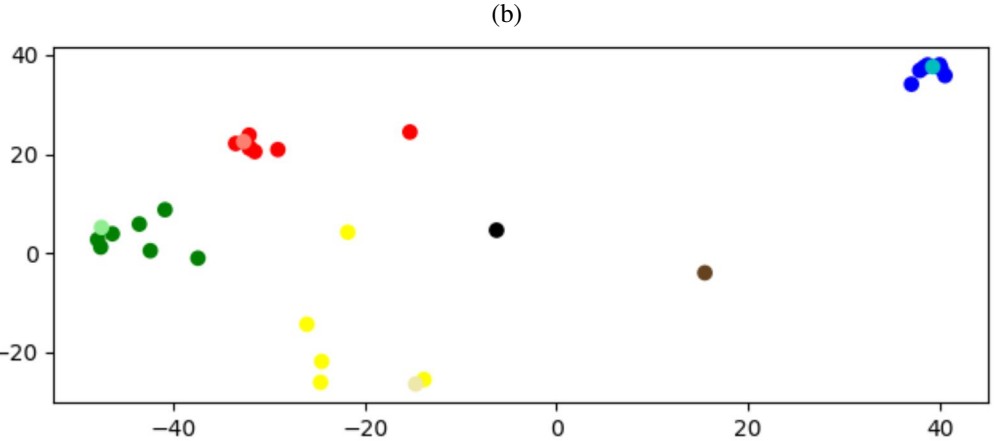

Figure 1: t-SNE Visualization of the semantic latent space. The representations of the sentences in (a) are plotted in (b). Multiple semantic related responses are close to each other and close to the corresponding prompt, while generic responses are far away.

For the second decoding step, we train an additional autoencoder. The decoder part is used for constructing the response sentence from the shared latent space. The autoencoder representation is split into two parts. One part correlates with the prompt and the other is independent of the prompt. This design models that some information of the response is related to the prompt, while other

information might be completely irrelevant (e.g. a follow up question). The encoder of the correlated part is shared with the canonical correlated feature extractor for responses, while the uncorrelated part is learned independently of the prompt and regularized using a normal distribution. The decoder is only trained on the autoencoder task. It simply recovers the sentence from its semantic latent representation, and it is not conditioned on the prompt. Since the semantics of the response and the decoding are learned separately, we could still perform beam search for the most probable sequence during inference but without the generic response problem.

We conduct experiments on two dialog datasets, and show that our model generates coherent and specific responses.

## 2 RELATED WORK

Several previous models also uses the idea of learning on sentence vector representations. (Luo et al., 2018) used two autoencoders to learn the semantic representations of inputs, and learned utterance-level dependency between those representations. (Gao et al., 2019) fused the autoencoder and seq2seq feature space, so that the distance and direction from a predicted response vector roughly matches the relevance and diversity. Those methods partially addresses this issue, but the problematic end-to-end MLE loss still plays an important role in both methods. In our work, we completely remove the end-to-end loss term (our autoencoder loss does not backpropagate through the prompt encoder), so the matching of the input and response is learned only on a shared semantic latent space. (Qiu et al., 2019) proposed a two-stage generation process, which predicts the average of the reference responses as an intermediate task, but it requires training data with multiple responses. (Kumar & Tsvetkov, 2019) explored predicting continious word vectors on the token level in seq2seq models instead of using softmax classifcation.

To tackle the generic response problem, (Li et al., 2016a) maximized mutual information in decoding or reranking. (Zhou et al., 2017) trained multiple response mechanisms to model diversity. (**?**) split the generation into segments and allow attention to attend to both the prompt and the response to improve diversity. Several works use explicit specificity metrics to manipulate the specificity of the responses, either using reinforcement learning(Li et al., 2016b), sample reweighting(Lison & Bibauw, 2017), or a gaussian mixture model (Zhang et al., 2018a). Frequency based metrics such as IDF are commonly used. (Ko et al., 2019) proposed using a specificity metric trained on discourse relation pair data.

Deep canonical correlation analysis (Andrew et al., 2013) has previously been used on various tasks including feature learning (Wang et al., 2015), caption retrieval (Yan & Mikolajczyk, 2015), multi-label classificaiton (Yeh et al., 2017), image cross-reconstruction (Chanda et al., 2016), and multilingual word similarity (Rotman et al., 2018). (Mallinar & Rosset, 2018) experimented on performing deep canonical correlation analysis (DCCA) on sequential data with a recurrent network.

## 3 OUR METHOD

Given example dialogue (Prompt, Response) pairs $(D_x, D_y)$ from open-domain dialogue datasets, our goal is to generate a coherent and non-generic response when given an unseen prompt. The structure of our model is depicted in Figure 2. It consists of three encoders; *Prompt Encoder $F_x$*, *Correlated Response Encoder $F_y$*, and *Uncorrelated Response Encoder $F_u$*. The final response is generated from a semantic latent vector via a *Decoder $G_y$*. During training all three encoders and the decoder are tuned. However, for inference/testing only the Prompt Encoder and the Decoder are utilized. We will now describe each of these components in more detail.

### 3.1 LEARNING THE CORRELATED SEMANTIC LATENT SPACE

Our primary aim is to learn a latent space where semantic related sentences (both prompts and responses) are close to each other. For this we employ canonical correlation analysis (CCA) (Hotelling, 1936) between the prompt and response pairs, which maximizes the correlation of the extracted features with the other sentence in the pair. Since semantically similar responses are likely to correspond to a similar set of prompts, semantic similar sentences will have similar represen-

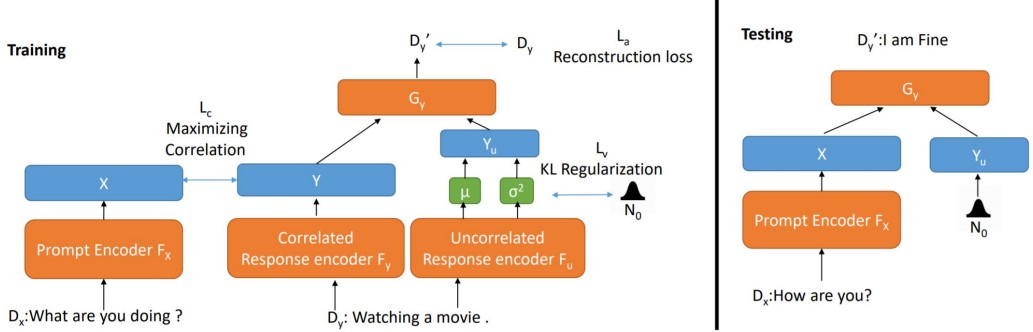

Figure 2: Architecture of our model.

tations in the CCA encoded space. Generic responses could be responses to a much larger set of prompts, so they will have very different representations in the latent space.

We use two recurrent neural networks $F_x, F_y$ as feature extractors to map prompts and responses into the shared featured space respectively. Using the definition of CCA, we maximize the total correlation of each dimension between $X = F_x(D_x), Y = F_y(D_y)$ as follows.

$$\max\left(\sum_{i=1}^{k} corr(X^i, Y^i)\right) = \max\left(\sum_{i=1}^{k} \frac{\sum_m (X_m^i - \bar{X}^i)(Y_m^i - \bar{Y}^i)}{\sqrt{\sum_m (X_m^i - \bar{X}^i)^2 \sum_m (Y_m^i - \bar{Y}^i)^2}}\right), \quad (1)$$

subject to the condition

$$\sum_m (X_m^i - \bar{X}^i)(X_m^j - \bar{X}^j) = \sum_m (Y_m^i - \bar{Y}^i)(Y_m^j - \bar{Y}^j) = 0 \quad (2)$$

for each $i, j$ pair. $i, j$ are the indices of the feature dimension, and $m$ is the index of the example pair in the batch, and $k$ is the number of feature dimensions. The condition ensures that the different dimensions in the representation are uncorrelated, to avoid redundant representations.

To make the two feature spaces $X$ and $Y$ shared, we add the following conditions to the mean and variance of both representations, inspired by (Yeh et al., 2017).

$$\bar{X}^i = \bar{Y}^i = 0 \quad (3)$$

$$\sum_m (X_m^i)^2 = \sum_m (Y_m^i)^2 = 1 \quad (4)$$

for all dimensions $i$.

When prompt $X$ and response $Y$ are perfectly correlated, and these two conditions perfectly hold, $X$ will be equal to $Y$, so this makes $X$ and $Y$ interchangeable during inference. This is desirable because we don't have access to Y during inference.

Using the two conditions, Equations 1, 2 becomes;

$$\max\left(\sum_{i=1}^{k} corr(X^i, Y^i)\right) = \max\left(\sum_{i=1}^{k} \sum_m X_m^i Y_m^i\right) = \min\left(\sum_{i=1}^{k} \sum_m (X_m^i - Y_m^i)^2\right), \quad (5)$$

subject to,

$$\sum_m X_m^i X_m^j = \sum_m Y_m^i Y_m^j = 0 \quad (6)$$

We can formulate the total CCA loss from equation 3 to equation 6 as,

$$L_c = \sum_{i=1}^{k} \left( \sum_m (X_m^i - Y_m^i)^2 + \lambda_1 \left( abs(\sum_m X_m^i) + abs(\sum_m Y_m^i) \right) \right.$$

$$\left. +\lambda_2 \left( abs(\sum_m (X_m^i)^2 - 1) + abs(\sum_m (Y_m^i)^2 - 1) \right) \right) + \lambda_3 \sum_{i,j}^{i \neq j} \left( abs(\sum_m X_m^i X_m^j) + abs(\sum_m Y_m^i Y_m^j) \right)$$

(7)

where $\lambda_1, \lambda_2, \lambda_3$ are tunable hyper-parameters.

## 3.2 GENERATING THE RESPONSE FROM THE SEMANTIC LATENT SPACE

Since $X$ and $Y$ are interchangeable in the semantic space, we directly use the features extracted from the prompt $X$ to approximate the response features $Y$. Now we want to generate the response sentence $D_y'$ from these response features. For this purpose, an additional autoencoder is trained on all the responses, simultaneously with the CCA. The autoencoder consists of encoder $F_y$, and decoder $G_y$, both of which are recurrent networks. The parameters of the encoder are shared with the semantic feature extractor of the responses. During inference, features extracted from the prompt $X$ is directly fed into the decoder to generate the response sentence.

$$D_y' = G_y(F_x(D_x))$$

(8)

We also experimented on adding domain discriminative adversarial training (Tzeng et al., 2017) between $X$ and $Y$, but it did not improve the results. This shows that our conditions (1),(3),(4) already makes the distribution of the two encoders sufficiently similar.

Generating sentences from a continuous space is known to produce ungrammatical text (Bowman et al., 2016). To address this issue, we randomly replace some of the autoencoder input word tokens by the unknown word token $\langle unk \rangle$. This serves three purposes. First, it makes the decoder more robust, and able to generate grammatical responses when there is noise in the decoder input. This is important because the decoder input is from a different encoder during inference. Second, it prevents the autoencoder from overfitting too early before the CCA objective converges. Finally, the learned hidden representation would be more high-level, includiing semantic knowledge. This is desirable since the same representation is also used for learning the CCA for the semantic latent space.

Note that, this is different from the word dropout used in (Bowman et al., 2016). They only replace decoder inputs, while we replace both encoder and decoder inputs, since we want to prevent overfitting and make the extracted features include more high level semantic information of the sentence instead of focusing on individual words.

## 3.3 CORRELATED AND UNCORRELATED REPRESENTATIONS

When encoding the response with $F_y$, the autoencoder and the CCA loss have conflicting objectives. The autoencoder task requires the representation to preserve all the information in the sentence for reconstruction. The CCA tasks aims to preserve only the information correlated with the prompt and discard all other irrelevant information. For example, a paraphrase pair are likely to be responses of the same set of prompts, so they should have the same representation under CCA objective, but the autoencoder objective forces the representations to be different, to enable reconstruction of the exact sentences. A response could also include a topic change, which makes part of the response completely irrelevant with the prompt, and the irrelevant information shouldn't be in the CCA representation.

To model this issue, we separate the autoencoder representations into the correlated part $Y$, which correlates with the prompt, and the uncorrelated part $Y_u$. The correlated part learns both the autoencoder task and the CCA task. The uncorrelated part is only trained for autoencoder reconstruction.

During training, $G_y$ learns to reconstruct from the concatenation of the correlated and uncorrelated representations. The reconstruction is trained using cross entropy loss.

$$D_y' = G_y([Y; Y_u]) = G_y([F_y(D_y); F_u(D_y)])$$

(9)

$$L_a = CrossEntropy(D_y, D'_y) \tag{10}$$

During testing, $G_y$ generates the response from the CCA semantic representation of the prompt and a vector $R$ representing the uncorrelated part of the response.

$$D'_y = G_y([F_x(D_x); R]) \tag{11}$$

By adding additional regularization to the uncorrelated representation $Y_u$ during training, we encourage a normal distribution with zero mean and unit variance. Hence during inference we can sample $R$ from this distribution or use a fixed prior to approximate $Y_u$.

The formulation of the regularization is the same as variational autoencoders (Kingma & Welling, 2013). An encoder recurrent network $F_u$ predicts a mean $\mu$ and variance $\sigma^2$ for each dimension, and $Y_u$ is sampled from that multivariate normal distribution. The predicted $\mu$ and $\sigma^2$ is regularized by the KL divergence with unit normal distribution.

$$L_v = \sum_i \sum_m ((\mu_m^i)^2 + (\sigma_m^i)^2 - log((\sigma_m^i)^2)) \tag{12}$$

With this regularization, $R$ can either be set to all zero mean, or be randomly drawn from a unit Gaussian distribution. We found that the generated sentence is insensitive to this choice, so the two ways generate exactly the same sentence more often than not. Despite the insensitivity, we calculated the ratio between the KL loss and the autoencoder reconstruction loss, and found that the ratio consistently increases during training, indicating that there is no posterior collapsing (Chen et al., 2017). We also found that adding the uncorrelated representation allows both the CCA loss and the autoencoder reconstruction loss to converge to a significantly lower value. Inspection on generated sentences showed that there are obvious improvement on coherence, at the cost of slightly increased grammatical errors.

To avoid introducing excessive noise while using $R$ as an approximation, we use a low dimension for the uncorrelated part. A high dimension results in worse performance in our experiments.

During training, the gradient of all loss terms are weighted summed and all parameters are updated together. The total loss is:

$$\mathcal{L} = \lambda_4 L_c + \lambda_5 L_a + \lambda_6 L_v \tag{13}$$

where $\lambda_4, \lambda_5, \lambda_6$ are hyper-parameters.

### 3.4 ATTENTION

One shortcoming of this model is that it does not have an attention mechanism, so it couldn't dynamically focus on different parts of the prompt during generation. We also experiment on a variant of our model with Luong et al. (2015)'s attenion. Similar to previous works, the key and value is from the RNN hidden state of the prompt encoder $F_x$, and the query is the hidden state of the response decoder $G_y$. To prevent nullifying the main purpose of our model design: removing end-to-end MLE training, we create a bottleneck to limit the end-to-end information flow before concatenating the attention output vector with the hidden state. The bottleneck is a fully connected layer that reduces the attention output vector into a low dimension.

## 4 EXPERIMENTS

### 4.1 DATASETS

We conduct experiments on two datasets: PersonaChat (Zhang et al., 2018b) and DailyDialog (Li et al., 2017). PersonaChat is a chit-chat dataset collected by crowdsourcing. We don't use the personas in the dataset since they are not related to our work. We use the ConvAI2 version, which contains 122,499 prompt-response pairs for training. We use 3,000 pairs for validation and 4,801 pairs for testing. DailyDialog is a collection of conversations in daily life for English learners. We remove those prompt-response pairs in the validation and test set that also appears in the training set, which resulted in about 30% of pairs removed in the test set. The final dataset has 76,052 pairs for training, 5,334 pairs for validation, and 4,738 pairs for testing.

### 4.2 Methodology

We compare with the vanilla Seq2seq model with attention (Luong et al., 2015) trained using MLE loss, decoded using beam search and nucleus sampling (Holtzman et al., 2019). We also compare with previous work MMI-anti (Li et al., 2016a) and SpaceFusion (Gao et al., 2019). MMI-anti is based on mutual information, it penalizes frequent responses with an anti-language model. Space-Fusion is a re cently proposed method which learns a fused common space representation of the Seq2seq dialogue generation task and the autoencoder task. We use the authors' code.

**Parameters:** We use 1 layer GRU for all encoders and decoders. The correlated representation size is 512, the uncorrelated representation size is 10. We implement $F_y$ and $F_u$ as different output dimensions of the same GRU. For compared methods we use hidden layer size 522. The word embedding dimension is 128. We use Adam (Kingma & Ba, 2015) optimizer with learning rate 0.001, $\beta_1 = 0.9, \beta_2 = 0.999$. Batch size is 64. $\{\lambda_1, \lambda_2, \lambda_3, \lambda_4, \lambda_5, \lambda_6\}$ is set to $\{3.9, 6.25, 0.05, 2, 2, 0.1\}$, they are tuned to make the conditions (2), (3), (4) enforced properly. The number of epochs trained is tuned on the validation set. Since there is no obvious automatic metric to evaluate the response quality, it is validated by human inspection on the generated sentences on the validation set for all models. Under this criterion, the MLE and MMI models are stopped considerably later than the minimum validation perplexity, since stopping earlier makes the responses more generic. For our model with attention, the attention bottleneck has dimension 10.

### 4.3 Automatic evaluation

Human evaluation is the only reliable way to evaluate the generated sentences, and the automatic metrics are only for reference. (Liu et al., 2016) showed that automatic metrics have low correlation with human judgements. However, since there are no good alternatives, those automatic metrics are still widely used. We use the following automatic evaluation metrics: (1) BLEU-1 and BLEU-2 (Papineni et al., 2002) (2) Embedding Average cosine similarity (Foltz et al., 1998) between generated and gold response. The sentence-level embedding are computed by averaging the embeddings of each word in the sentence. This metric reflects the coherence of the response. (3) dist-1 and dist-2 (Li et al., 2016a), which evaluates the diversity of the generated responses. They respectively calculate the count of distinct unigrams and bigrams, divided by the total number of words in all responses. Those metrics are also used in (Zhang et al., 2018a; Xu et al., 2018; Gu et al., 2019; Qiu et al., 2019).

Table 1 shows the results. Our model outperform baselines on BLEU and embedding similarity. In terms of dist score, our model is not higher than MMI, since the responses our model generate are much longer.

### 4.4 Human evaluation

We use crowdsourcing for human evaluation. Following the standards of (Shao et al., 2017; Liu et al., 2018; Zhang et al., 2018a; Qiu et al., 2019), for each model, we let 3 workers label 500 pairs of randomly sampled data. Our evaluation involved a pool of around 400 distinct workers in total. We ask them to rate the specificity and coherence of each response, on a Likert scale from 0 to 3. We report the average of all annotations for both metrics. Since a good response has to be both coherent and non-generic, we also report the useful information (UI) score, which is multiplying specificity multiplied by coherence for each sentence, and then take average over all sentences.

On PersonaChat dataset, SpaceFusion and sampling could generate very specific responses, but they do not improve coherence. Our model generates the most coherent responses, and the specificity score outperforms Seq2Seq and MMI. On DailyDailog dataset, sampling scores highest on specificity, but the responses are not coherent. Our model with attention greatly improves specificity compared to other baselines, while maintaining similar level of coherence. Adding attention to our model improves both coherence and specificity on the DailyDialog dataset, but harms specificity on the PersonaChat dataset. Examples of generated responses are shown in the appendix. Our models performs best on UI score for both datasets. The improvements are significant.

We also evaluated the grammatical correctness of our model on PersonaChat. About 11% of sentences contain major grammatical errors that cause difficulty understanding of the sentence. 18% contain minor errors that don't affect the understanding of the sentence. 71% of the sentences are

grammatically correct. This shows that most of the responses of our model are acceptable by human, and comprehensibility is not a major problem while we generate responses from a continuous space. Example model responses are shown in appendix A.

| Dataset | Model | automatic metrics | | | | | human evaluation | | |
|---|---|---|---|---|---|---|---|---|---|
| | | Bleu-1 | Bleu-2 | Sim | Dist-1 | Dist-2 | Coh. | Spec. | UI |
| PersonaChat | MLE+beam search | 0.146 | 0.0640 | 0.854 | 0.0128 | 0.0397 | 1.09 | 0.98 | 1.43 |
| | MLE+sampling | 0.148 | 0.0605 | 0.851 | 0.0313 | 0.131 | 1.05 | 1.39 | 1.65 |
| | MMI | 0.124 | 0.0551 | 0.838 | **0.0517** | 0.217 | 1.18 | 1.08 | 1.54 |
| | SpaceFusion | 0.176 | 0.0715 | 0.855 | 0.0233 | 0.0766 | 1.05 | **1.66** | 1.84 |
| | Ours | **0.191** | **0.0746** | **0.871** | 0.0363 | 0.179 | 1.29 | 1.35 | **1.97** |
| | Ours+Attention | 0.182 | 0.0712 | 0.868 | 0.0360 | 0.171 | **1.32** | 1.22 | 1.76 |
| | Ours w/o Uncorrelated | 0.179 | 0.0729 | 0.866 | 0.0305 | 0.127 | 0.96 | 1.40 | 1.46 |
| | Ours w/o Denoising | 0.167 | 0.0556 | 0.868 | 0.0332 | **0.252** | - | - | - |
| DailyDialog | MLE+beam search | 0.100 | 0.0394 | 0.763 | 0.0440 | 0.152 | 1.25 | 0.73 | 0.84 |
| | MLE+sampling | 0.142 | 0.0413 | 0.790 | 0.0620 | **0.389** | 0.81 | **1.56** | 1.18 |
| | MMI | 0.0939 | 0.0369 | 0.764 | **0.0697** | 0.270 | **1.29** | 0.89 | 1.09 |
| | SpaceFusion | 0.146 | **0.0595** | 0.792 | 0.0531 | 0.216 | 1.21 | 1.08 | 1.24 |
| | Ours | 0.170 | 0.0575 | **0.807** | 0.0457 | 0.191 | 1.18 | 1.45 | 1.71 |
| | Ours+Attention | **0.171** | 0.0558 | 0.805 | 0.0530 | 0.213 | 1.24 | 1.51 | **1.74** |

Table 1: Result comparison of all models on PersonaChat and DailyDialog datasets using different automatic and human evaluation metrics. Our model generates specific and coherent responses.

## 4.5 ABLATION STUDY

We compare our full model with two variants and test the contribution of different parts in our model. We use the PersonaChat dataset for this experiment. The **w/o Uncorrelated part** model does not have the representation $Y_u$, the autoencoder reconstruction is solely based on $Y$, which also learns the CCA task. In the **w/o Denoising** model, we do not replace random words with $\langle unk \rangle$ in the autoencoder input. The results are in Table 1.

Without the uncorrelated part, all automatic metrics decrease. Human inspections of sentences show that there is a obvious decrease on coherence, showing that allowing uncorrelated information is important for the learning the correlation between the prompt-response pairs. Without denoising, the generated sentences contains many grammatical errors. All automatic metrics decreased except dist-2, which is caused by the ungrammaticality. Since the sentences are obviously unacceptable by humans, we did not perform human evaluation. This shows that denoising is critical for our model to generate grammatical responses.

## 4.6 VISULIZING THE SEMANTIC SPACE

In order to verify that the shared latent space successfully encodes semantic information, we visualize the representaions of some sentences in Figure 1. The dimension reduction is performed using t-SNE (van der Maaten, 2009) trained on 1,000 prompt representations and 1,000 response representations in the test set.

The light red point is the latent representation of the sentence *"what instruments do you play ?"* encoded by the prompt encoder $F_x$. The seven dark red points are possible responses encoded by the response encoder $F_y$, such as *"i practice the piano every day ."*, *"i am learning the guitar ."* Similarly, the light and dark blue, green, and yellow points shows multiple possible responses to other three different questions.

We can see that those semantically related responses to the same question are clustered, showing that the latent space is indeed able to capture semantic information. The questions' representation is close to the cluster of their corresponding responses, demonstrating that our model has successfully learned from the collection of semantically similar possible responses.

We also visualize *"i don't know ."* in black, and the most frequent generic response generated by the Seq2Seq model trained on PersonaChat, *"what do you do for a living ?"* in brown, using the

response encoder $F_y$. Those generic responses are much farther away from the question than specific responses, thus they are unlikely to be generated.

## 5 CONCLUSION

In this paper, we proposed a novel open domain dialogue generation framework that first predicts a semantic sentence vector, and then reconstruct it into a sentence. Our model jointly learns an denoising autoencoder and a semantic latent space using CCA. The autoencoder representation consist of two parts, one correlates with the prompt, and one is regularized to follow a unit normal distribution. This design enables learning from the semantic relatedness of different responses. Our model could generate coherent responses, and doesn't suffer from the generic response problem unlike previous approaches. Evaluation on open domain dialog datasets concretely demonstrates the efficacy of our model.

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

## A  EXAMPLE MODEL RESPONSES

| PersonaChat | |
|---|---|
| **Prompt** | wow i love to see traditional dance . |
| **MLE+beam search** **MMI** **Spacefusion** **Ours** | do you have a favorite color ? yes , i love the color blue . i am not sure . i am a college student . its fun . i've always wanted to go for a dance class . |
| **Prompt** | do you go to school at all |
| **MLE+beam search** **MMI** **Spacefusion** **Ours** | i do not have a job but i want to be a doctor sometimes , but i do like photography . i do not . i am a college student . i don't . i'm studying to go to college . |
| **Prompt** | i live in nebraska . i don't think i have ever been to ohio before . |
| **MLE+beam search** **MMI** **Spacefusion** **Ours** | oh wow . what do you do for a living ? oh do you do volunteer work oh wow . i love to go to the beach . wow , i have been there . do you have a favorite place ? |
| DailyDialog | |
| **Prompt** | Can you operate computers skillfully ? |
| **MLE+beam search** **MMI** **Spacefusion** **Ours+Attention** | Yes , I am . Yes , I have a special job . Yes , I can . I have any other Word 2003 , and I can live in other areas . No , I am not familiar with both Java and C Programming Languages . |
| **Prompt** | Exporters must ensure that their product satisfies customers ' needs , wants and likes . |
| **MLE+beam search** **MMI** **Spacefusion** **Ours+Attention** | Do you have any other questions ? We have to buy a new set . I am calling to see you . But I've got a new customer here . But I don't want to be aware of their house . That means that we have a commission about selling electronics . The prices are perfect for you to promote the commission and the prices are cheaper . |
| **Prompt** | It closes at 7:00 on Sundays . |
| **MLE+beam search** **MMI** **Spacefusion** **Ours+Attention** | You're welcome , I can help you . How do you get that ? Nice to see you . The alarm isn't a fool . Hold on , I have some other classes available . |

