# OpenReview forum: "Generating Dialogue Responses From A Semantic Latent Space"
_ICLR.cc/2020/Conference — Reject_

### Official Review · AnonReviewer1 · 2019-10-22
**Official Blind Review #1**

**Rating:** 3

**Review:**

This paper proposes a method for generating dialogue responses that are not generic. The authors propose maximizing correlation between latent representations of responses and prompts, such that the response is encouraged to contain information relevant to the prompt. The method consists of two parts, the first part predicts a latent representation of the response while the second part decodes a response from the latent representation.

The idea is conceptually intuitive, however I have some questions regarding the details:
- It's surprising that this works at all, since the inference representations seem to come from an entirely different distribution than in training (e.g. Figure 2)
- Given that this works, I'd like to see an ablation in which Y is removed (e.g. we go from X and Yu to Gy), so as to evaluate the utility of the "correlated response encoder". This is effectively what's happening during inference anyway.
- I am skeptical about the early stopping criteria, which involve human evaluation. I also don't find the justification convincing ("there is no obvious automatic metric") --- why don't we just use the automatic metrics listed in 4.3?
- Can the authors please comment on the choice of evaluation? It seems like the evaluation metrics are different from what's typically used (e.g. see http://convai.io for PersonaChat). Why don't the authors use these metrics instead? This, combined with the early stopping criterion, make it difficult to evaluate the effectiveness of the proposed method in light of prior work.

In terms of the writing, I think the authors could do with more polish:
- there are many generic claims which can be more detailed (e.g. abstract: enables our model to view semantically related responses collectively; achieving [...] better coherence)
- some terms are not clearly defined (e.g. semantics, one-to-many task vs. one-to-one task, canonical correlated feature extractor)
- the introduction is overly verbose, much of it should be in the methodology section. there should be an overview of experimental results and analysis in the intro.
- In 3.3, it doesn't seem like you "enforce" a normal distribution. If I understand correctly, you use KL to encourage a normal distribution.

**Experience Assessment:**

I have published one or two papers in this area.

**Review Assessment: Checking Correctness Of Derivations And Theory:**

I did not assess the derivations or theory.

**Review Assessment: Checking Correctness Of Experiments:**

I assessed the sensibility of the experiments.

**Review Assessment: Thoroughness In Paper Reading:**

I read the paper at least twice and used my best judgement in assessing the paper.

---

> ### Author Response · Authors · 2019-11-07
> **Response to reviewer 1**
>
> Thank you for the review. We respond to your comments below, and we will revise the paper to clarify those issues.
>
> - Given that this works, I'd like to see an ablation in which Y is removed (e.g. we go from X and Yu to Gy), so as to evaluate the utility of the "correlated response encoder". This is effectively what's happening during inference anyway.
>
> If we remove Y and train X in an end-to-end way with Yu and Gy, the model becomes similar to CVAE[1] but with inputs partially removed, so it would not outperform CVAE. It was shown that CVAE produces quite generic responses.[2]
> [1] Tiancheng Zhao, Ran Zhao and Maxine Eskenazi  Learning Discourse-level Diversity for Neural Dialog Models using Conditional Variational Autoencoders ACL 2017
> [2] Jun Gao , Wei Bi, Xiaojiang Liu, Junhui Li, Guodong Zhou , Shuming Shi, A Discrete CVAE for Response Generation on Short-Text Conversation, EMNLP 2019
>
>
>  - I am skeptical about the early stopping criteria, which involve human evaluation. I also don't find the justification convincing ("there is no obvious automatic metric") --- why don't we just use the automatic metrics listed in 4.3?
> -Can the authors please comment on the choice of evaluation? It seems like the evaluation metrics are different from what's typically used. Why don't the authors use these metrics instead?
>
> Human evaluation is the only reliable method to evaluate this task, and the results clearly showed our method is effective.
>
> Automatic evaluation is only for reference. [3] showed that commonly used automatic metrics for dialog generation have very low correlation with.human judgement. However, since there are no good alternatives, those metrics are still widely used in literature. The automatic metrics used in our work are also used in [4]~[7] and [1] , [2]. Since those metrics in [3] have very low correlation with human evaluation, they are not suitable for stopping criteria. Distinct also couldn’t be used since it monotonically increases as the model overfits.
>
> Because our model is optimized on the latent space, instead of trained as a probabilistic model that minimizes perplexity, the perplexity of the gold responses in our model is very large (10^3~10^4). Nontheless, our model outperforms baselines on human evaluation. This is why we don’t use perplexity for evaluation as in ConvAI challenge. The other two metrics in the challenge are less often seen in the literature. F1 is similar to the BLEU we used. Hits@1 are mainly for ranking models and requires candidate responses.
>
> [3] Chia-Wei Liu, Ryan Lowe, Iulian V. Serban, Michael Noseworthy, Laurent Charlin, Joelle Pineau,How NOT To Evaluate Your Dialogue System: An Empirical Study of Unsupervised Evaluation Metrics for Dialogue Response Generation,EMNLP2016
> [4] Ruqing Zhang, Jiafeng Guo, Yixing Fan, YanyanLan, Jun Xu, and Xueqi Cheng. Learning to control the specificity in neural response generation. ACL 2018
> [5] Xinnuo Xu, Ondřej Dušek, Ioannis Konstas, Verena Rieser , Better Conversations by Modeling, Filtering, and Optimizing for Coherence and Diversity, EMNLP 2018
> [6] Xiaodong Gu, Kyunghyun Cho, Jung-Woo Ha, Sunghun Kim, DialogWAE: Multimodal Response Generation with Conditional Wasserstein Auto-Encoder Xiaodong Gu, Kyunghyun Cho, Jung-Woo Ha, Sunghun Kim ICLR 2019
> [7] Lisong Qiu, Juntao Li, Wei Bi, Dongyan Zhao, Rui Yan Are training samples correlated? learning to generate dialogue responses with multiple references. ACL, 2019
>
> - It's surprising that this works at all, since the inference representations seem to come from an entirely different distribution than in training (e.g. Figure 2)
>
> It would probably be more intuitive when comparing with domain discriminative training. [8]
> We mentioned in section 3.2 that “We also experimented on adding adversarial training between X and Y , but it did not improve the results.”, showing our that our constraints (1)(3)(4) already makes the two distributions sufficiently similar.
> [8] Eric Tzeng, Judy Hoffman, Kate Saenko, Trevor Darrell, Adversarial Discriminative Domain Adaptation, CVPR 2017

---

### Official Review · AnonReviewer3 · 2019-10-23
**Official Blind Review #3**

**Rating:** 3

**Review:**

[Summary]
This paper proposes a dialogue generation model that learns a semantic latent space. In this paper, the authors tackle the generic response issue. The proposed model consists of three (prompt/correlated response/uncorrelated response) encoders and a decoder. The authors add an objective that maximizes the correlation between prompt and response features to the existing autoencoder structure, which prevents the generation of generic responses.

[Comments]
The paper is well-organized and clearly written but has some weakness:
- There seems to be a need for more detail on comparisons of papers that previously tackle the generic response problem.
- In addition to the proposed main architecture, there are so many things combined (ex. denoising, attention). It seems like it's doing well with additional things rather than the model structure. In fact, the results of the model without denoising in the ablation study show poor performance.
- The test pool for human-evaluation is too small (only 3 workers). It seems necessary to perform the human-evaluation on a larger number of participants. I think that it is a very important issue since the generic response addressed as the main problem in this paper can only be evaluated by human-evaluation.

[Questions]
- What are the criteria for specificity and coherence used in human-evaluation? If there are some criteria provided to the participants, it would be better to include a description together.
- Can you provide an intuitive or theoretical explanation of denoising (to add a <unk>)?

**Experience Assessment:**

I have read many papers in this area.

**Review Assessment: Checking Correctness Of Derivations And Theory:**

I assessed the sensibility of the derivations and theory.

**Review Assessment: Checking Correctness Of Experiments:**

I assessed the sensibility of the experiments.

**Review Assessment: Thoroughness In Paper Reading:**

I read the paper at least twice and used my best judgement in assessing the paper.

---

> ### Author Response · Authors · 2019-11-07
> **Response to reviewer 3**
>
> Thank you for the review. We respond to the comments in your review below, and we will revise the paper to clarify those issues.
>
>  - There seems to be a need for more detail on comparisons of papers that previously tackle the generic response problem.
>
> We will revise the paper and add more discussion on previous work about the generic response problem.
>
>  - In addition to the proposed main architecture, there are so many things combined (ex. denoising, attention). It seems like it's doing well with additional things rather than the model structure. In fact, the results of the model without denoising in the ablation study show poor performance.
>
> We do not agree that using attention and denoising is a “weakness”. Most dialog generation models use attention. Also as shown in table 1, our model without attention still outperform other compared methods in human evaluation.
> Since other compared models could not be improved similarly by using denoising, it shouldn’t be considered an “additional thing”, but a critical part of our model instead.
>
>  - The test pool for human-evaluation is too small (only 3 workers). It seems necessary to perform the human-evaluation on a larger number of participants. I think that it is a very important issue since the generic response addressed as the main problem in this paper can only be evaluated by human-evaluation.
>
> We follow the standards of [1]~[4] to use 3 distinct workers per prompt-response pair. (Common range in literature is 3~5 workers)
> Also we evaluate 500 pairs of sentences for each model, which is on the high side compared to previous work. (Common range is 100~500 pairs)
> Note that overall our evaluation involved a pool of around 400 distinct crowd-workers.
> [1]Louis Shao, Stephan Gouws, Denny Britz, Anna Goldie, Brian Strope, Ray Kurzwei, Generating High-Quality and Informative Conversation Responses with Sequence-to-Sequence Models
> EMNLP2017
> [2]Yahui Liu, Wei Bi, Jun Gao, Xiaojiang Liu, Jian Yao, Shuming Shi, Towards Less Generic Responses in Neural Conversation Models: A Statistical Re-weighting Method EMNLP2018
> [3]Ruqing Zhang, Jiafeng Guo, Yixing Fan, YanyanLan, Jun Xu, and Xueqi Cheng. Learning to control the specificity in neural response generation. ACL 2018
> [4] Lisong Qiu, Juntao Li, Wei Bi, Dongyan Zhao, Rui Yan.  Are training samples correlated? learning to generate dialogue responses with multiple references. ACL, 2019
>
>  - What are the criteria for specificity and coherence used in human-evaluation? If there are some criteria provided to the participants, it would be better to include a description together.
>
> For specificity, the workers are asked to rate the utterances according to “how specific and detailed the response is”. They are instructed that “A specific sentence provides information that relates uniquely to a particular subject. If the response contains both generic and specific parts, rate the most specific part”. They are also provided example responses of each score 0~3 for that dataset.
>
> For coherence, the workers are asked to rate “if the response is relevant to the prompt“. They are instructed that “A response is relevant if part of it discusses a similar topic with part of to the prompt, answers a question asked in the prompt, or giving a suitable response to a situation. If other parts of the response are about a different topic, it is still considered coherent”. They are also provided example prompt-response pairs for each score 0~3 for that dataset.
>
>
> - Can you provide an intuitive or theoretical explanation of denoising (to add a <unk>)?
>
> The <unk> makes the decoder more robust, and able to generate grammatical responses when there is noise in the decoder input. This is important because the decoder input is from a different encoder during inference.
> Also without denoising, the autoencoder will severely overfit, which is a main reason why it show poor performance
> Finally, the learned hidden representation would be more high-level, includiing semantic knowledge. This is desirable since the same representation is also used for learning the CCA for the semantic latent space..

---

### Official Review · AnonReviewer2 · 2019-10-26
**Official Blind Review #2**

**Rating:** 3

**Review:**

1. Summary: The authors proposed to alleviate the generic response problem in open-domain dialog generation by generating a response to a prompt from a semantic latent vector. This vector needs to be located close to the latent vector of the corresponding prompt. To this end, they employ canonical correlation analysis and auto-encoder to learn the mapping from a sequence of text to a semantic latent vector. To model the variations and topic shifts that may happen in responses, they also use a separate intermediate vector in the auto-encoder of generating responses.
2. Overall assessment: While this paper is quite fun to read, it is not innovative enough and it lacks some critical experiments and error analysis to be accepted this time. I'll elaborate on these problems in my comments below.
3. Strengths:
3.1 This paper is well written. The motivation and the main idea are well explained and pretty easy to understand. Although there are some details missing, it doesn't prevent readers from understanding and enjoying this paper.
3.2 The use of human evaluation is a great plus. Subtle characteristics of text, such as readability, coherence, and specificity cannot be well justified by automatic evaluation. Human evaluation is a must for these aspects. It's great to see the authors include this in this paper.
4. Weakness and questions:
4.1 It seems the model in this paper can be connected to adversarial learning from some angles. It would be great to see such analysis in this paper.
4.2 The experiments in this paper do not look thorough enough. There are many more things should be included, such as error analysis, comparisons over different variations of the proposed model and so on. What if we remove $Y_u$ in both training and inference but keep the overall model the same? How important is $Y_u$ and what's its effect? Many such questions haven't been answered in this paper.
4.3 Embedding average cosine similarity seems to be too simple to be used for evaluation as it omits the contextual information and semantic meaning of a sentence.
4.4 It would be interesting to see the Dist-1 and Dist-2 scores of the gold standard responses. It's a good reference to let us know ho diversified the real responses are.
4.5 I'd like to see some explanation of how the raters rate specificity and coherence. What are the standards they use? What are the instructions the author give to them?
4.6 Are improvements significant in Table 1? It seems the proposed model is not performing very stably over different metrics. Some anlysis on this would be great.

**Experience Assessment:**

I have read many papers in this area.

**Review Assessment: Checking Correctness Of Derivations And Theory:**

I assessed the sensibility of the derivations and theory.

**Review Assessment: Checking Correctness Of Experiments:**

I assessed the sensibility of the experiments.

**Review Assessment: Thoroughness In Paper Reading:**

I read the paper at least twice and used my best judgement in assessing the paper.

---

> ### Author Response · Authors · 2019-11-07
> **Response to reviewer 2**
>
> Thank you for the review. We respond to the comments in your review below, and we will revise the paper to clarify those issues.
>
> >4.1 It seems the model in this paper can be connected to adversarial learning from some angles. It would be great to see such analysis in this paper.
>
> We agree that the correlation objective can be connected to domain discriminative training, we will discuss it in the paper.
>
> >4.2 The experiments in this paper do not look thorough enough. There are many more things should be included, such as error analysis, comparisons over different variations of the proposed model and so on. What if we remove Yu in both training and inference but keep the overall model the same? How important is Yu and what's its effect? Many such questions haven't been answered in this paper.
>
> Removing Yu was already discussed in section 4.5: “Without the uncorrelated part, all automatic metrics decrease. Human inspections of sentences show that there is a obvious decrease on coherence, showing that allowing uncorrelated information is important for the learning the correlation between the prompt-response pairs.” The results were shown in table 1, the “Ours w/o Uncorrelated” model.
>
> >4.3 Embedding average cosine similarity seems to be too simple to be used for evaluation as it omits the contextual information and semantic meaning of a sentence.
>
> Human evaluation is the only reliable method to evaluate this task, and the results clearly showed our method is effective.
> Automatic evaluation is only for reference. [1] showed that commonly used automatic metrics for dialog generation have very low correlation with.human judgement. However, since there are no good alternatives, those metrics are still widely used in literature.
> The embedding average cosine similarity is also used in [2]~ [5].
> [1] Chia-Wei Liu, Ryan Lowe, Iulian V. Serban, Michael Noseworthy, Laurent Charlin, Joelle Pineau,How NOT To Evaluate Your Dialogue System: An Empirical Study of Unsupervised Evaluation Metrics for Dialogue Response Generation, EMNLP2016
> [2] Ruqing Zhang, Jiafeng Guo, Yixing Fan, YanyanLan, Jun Xu, and Xueqi Cheng. Learning to control the specificity in neural response generation. ACL 2018
> [3] Xiaodong Gu, Kyunghyun Cho, Jung-Woo Ha, Sunghun Kim, DialogWAE: Multimodal Response Generation with Conditional Wasserstein Auto-Encoder ICLR 2019
> [4] Xinnuo Xu, Ondřej Dušek, Ioannis Konstas, Verena Rieser , Better Conversations by Modeling, Filtering, and Optimizing for Coherence and Diversity, EMNLP 2018
> [5] Lisong Qiu, Juntao Li, Wei Bi, Dongyan Zhao, Rui Yan.  Are training samples correlated? learning to generate dialogue responses with multiple references. ACL, 2019
>
> >4.4 It would be interesting to see the Dist-1 and Dist-2 scores of the gold standard responses. It's a good reference to let us know how diversified the real responses are.
>
> PersonaChat
> Dist-1 = 0.0686
> Dist-2 = 0.379
> DailyDialog
> Dist-1 = 0.0922
> Dist-2 = 0.437
>
>
> >4.5 I'd like to see some explanation of how the raters rate specificity and coherence. What are the standards they use? What are the instructions the author give to them?
>
> For specificity, the workers are asked to rate the utterances according to “how specific and detailed the response is”. They are instructed that “A specific sentence provides information that relates uniquely to a particular subject. If the response contains both generic and specific parts, rate the most specific part”. They are also provided example responses of each score 0~3 for that dataset.
>
> For coherence, the workers are asked to rate “if the response is relevant to the prompt“. They are instructed that “A response is relevant if part of it discusses a similar topic with part of to the prompt, answers a question asked in the prompt, or giving a suitable response to a situation. If other parts of the response are about a different topic, it is still considered coherent”. They are also provided example prompt-response pairs for each score 0~3 for that dataset.
>
>
> >4.6 Are improvements significant in Table 1? It seems the proposed model is not performing very stably over different metrics. Some analysis on this would be great.
>
> The improvements of human evaluation are significant.
> Due to the deficiencies of automatic metrics, significance analysis on them would not be very meaningful.
>
>
> >-While this paper is quite fun to read, it is not innovative enough
>
> While our model is an application of existing machine learning algorithms and techniques, the major innovation of our work is learning a conditional language generation task on a sentence level latent space, instead of the maximizing probability on vocabulary for each word, which we believe is a novel concept to the NLP community.

---

### Author Response · Authors · 2020-10-18
**Camera-ready Version**

EMNLP 2020
https://arxiv.org/abs/2010.01658

---

### Decision · Program_Chairs · 2019-12-19

**Decision:**

Reject

**Comment:**

This paper proposes a response generation approach that aims to tackle the generic response problem. The approach is learning a latent semantic space by maximizing the correlation between features extracted from prompts and responses. The reviewers were concerned about the lack of comparison with previous papers tackling the same problem, and did not change their decision (i.e., were not convinced) even after the rebuttal. Hence, I suggest a reject for this paper.